# Harnessing Novel Reduced Graphene Oxide-Based Aerogel for Efficient Organic Contaminant and Heavy Metal Removal in Aqueous Environments

**DOI:** 10.3390/nano14211708

**Published:** 2024-10-26

**Authors:** Sunith B. Madduri, Raghava R. Kommalapati

**Affiliations:** 1Center for Energy and Environmental Sustainability, Prairie View A&M University, Prairie View, TX 77446, USA; sbmadduri@pvamu.edu; 2Department of Civil and Environmental Engineering, Prairie View A&M University, Prairie View, TX 77446, USA

**Keywords:** cysteamine, NMR, methyl orange, nickel ion, chemisorption, wastewater treatment

## Abstract

Ensuring clean water sources is pivotal for sustainable development and the well-being of communities worldwide. This study represents a pioneering effort in water purification, exploring an innovative approach utilizing modified reduced graphene oxide (rGO) aerogels. These advanced materials promise to revolutionize environmental remediation efforts, specifically removing organic contaminants from aqueous solutions. The study investigates the exceptional adsorption properties of rGO-aerogel, enhanced with cysteamine, to understand its efficacy in addressing water pollution challenges. The characterization methods utilized encompass various analytical techniques, including FE-SEM, BET, FTIR, TGA, DSC, XPS, NMR, and elemental analysis. These analyses provide valuable insights into the material’s structural modifications and surface chemistry. The research comprehensively explores the intricacies of adsorption kinetics, equilibrium, and isothermal study to unravel the underlying mechanisms governing contaminant removal. MO and Ni^2+^ exhibited adsorption of 542.6 and 150.6 mg g^−1^, respectively, at 25 °C. Ni^2+^ has unveiled the highest removal at pH 5, and MO has shown high removal in a wide pH range (pH 4–7). Both contaminants have shown fast adsorption kinetic performance on an rGO-aerogel surface. This study aims to identify the synergistic effect of cysteamine and rGO in aerogel formation to remove heavy metals and organic contaminants. These findings mark a significant stride in advancing sustainable water-treatment methods and pioneering in synthesizing innovative materials with versatile applications in environmental contexts, offering a potential solution to the global water pollution crisis.

## 1. Introduction

Water pollution remains one of the most pressing environmental challenges of our time, posing significant threats to ecosystems, human health, and the availability of clean water worldwide [1]. Heavy metals and organic dyes are among the many contaminants that harm water bodies, but they should be particularly avoided because of their innate toxicity, environmental durability, and capacity for bioaccumulation [2,3]. This study explores the removal of methyl orange (MO), a popular organic dye, and nickel ion (Ni^2+^), a common heavy metal contaminant from wastewater. It also looks at the origins, health effects, regulatory limitations, and wider implications for water quality [4,5]. It is crucial to note that both Ni^2+^ and MO have the potential to bioaccumulate, meaning they can build up in the tissues of organisms over time, leading to higher concentrations at each level of the food chain [6,7,8].

A heavy metal, nickel, enters aquatic habitats as a result of numerous human activities. The main causes of Ni^2+^ pollution are industrial discharges from mining activities and manufacturing processes, and the inappropriate disposal of items containing Ni^2+^. Major contributors are the mining, battery electroplating, and stainless-steel manufacturing industries. When releasesd to water bodies, Ni^2+^ does not break down biologically; therefore, it can linger and build up over time [6,9]. The ingestion of contaminated water or other routes can expose humans to Ni^2+^, which can cause serious health problems, such as skin rashes, cardiovascular disorders, and respiratory disorders. Long-term exposure to Ni^2+^ can also harm vital organs, including the liver and kidneys, and induce gastrointestinal distress [10,11]. Ni^2+^ poisoning has similarly detrimental effects on aquatic creatures’ growth, reproduction, and survival, upsetting food chains and reducing biodiversity. As a result, the ecological effects are equally severe [3]. In parallel, the printing, paper, leather, and textile industries make substantial use of organic dyes such as MO [8]. These dyes, which are distinguished by their vivid hues, become extremely noticeable even in minute amounts, adding to aesthetic pollution and presenting serious environmental risks [12,13]. MO is an azo dye, which is particularly concerning due to its potential health risks. As a possible carcinogen, it can trigger allergic reactions, skin irritation, and respiratory problems. One major cause of environmental contamination is the dyeing and printing industries’ discharge of untreated or insufficiently treated effluent [14,15].

In order to reduce the negative consequences of these pollutants, the regulatory framework for their management is essential. To protect the public’s health, the U.S. Environmental Protection Agency (EPA) set a maximum contamination level (MCL) for Ni^2+^ in drinking water at 0.1 mg L^−1^ [16]. The administrative laws under the Clean Water Act require industries to treat their effluents to eliminate hazardous compounds, including dyes, before discharge [17], even if particular limits for MO are not explicitly stated. The harmful effects of organic colors and heavy metals in water bodies are compounded. Because of their harmful impacts on different species, these pollutants contribute to the loss of biodiversity by causing population numbers and diversity to decline [1,4]. The degradation of water quality make water resources unfit for human consumption, recreational use, and maintenance of aquatic life [2].

Since ancient times, adsorption has been used for separation and purification, and it has been more popular than other water treatment techniques [18,19]. This process, where gas or liquid molecules adhere to a solid substrate’s surface, involves the preferential transfer and accumulation of substances from a fluid phase onto a solid surface [20,21,22]. Van der Waals forces and electrostatic interactions between the adsorbate molecules and the atoms of the adsorbent surface drive physical adsorption [23]. Suitable adsorbents have a high surface area and a higher number of polar functional groups. The surface area provides capacity, and polarity determines affinity. Micropores in adsorbents influence accessibility to internal surfaces. Pore size distribution can be engineered for selective separations [24,25]. Polar adsorbents like aluminosilicates are hydrophilic and have a liking for water. Nonpolar carbonaceous and most polymer adsorbents are hydrophobic with strong interactions with organic contaminations [26,27]. In our work, rGO-aerogel has revealed a mixture of hydrophobic and hydrophilic behaviors that aided in removing cations with ionic interactions and organic contaminants with Van der Waal interactions. Adsorption is one of the most effective methods for purifying water because locally available adsorbents are inexpensive and easy to dispose of. Moreover, natural adsorbents are the most promising due to their economical and practical advantages over traditional methods for removing organics like pesticides [28,29,30].

Graphene oxide (GO) is synthesized by oxidizing graphene, introducing oxygen-containing functional groups, and increasing the distance between layers [31]. In contrast, rGO is derived from graphene oxide using chemical, thermal, or other reduction processes that remove oxygen and restore electrical conductivity. While GO is an insulating, highly oxidized material, rGO has decreased oxygen content and properties closer to pristine graphene due to the removal of oxygen groups during reduction [32]. GO is the oxidized form of graphite, while rGO is the reduced form of graphene oxide with lower oxygen levels and restored graphitic properties [33]. rGO, with its graphene sheet-like structure and residual oxygen functionalities like hydroxyl, carbonyl, and carboxyl groups, mainly at the edge and defect sites, can be used in catalysts, water purification, membranes, electronic devices, etc. [34,35,36]. The restoration of the sp^2^ network leads to enhanced properties compared to GO [37].

After chemical treatment with cysteamine, rGO is further modified to form rGO-aerogel. Aerogels are a unique class of materials known for their extremely low density and high porosity, often referred to as “frozen smoke” due to their translucent appearance and lightweight nature, holding immense potential for various applications. The rGO-aerogel, in particular, is constructed from a network of graphene sheets, which impart exceptional properties to the material [34,36]. In the formation process, the rGO-aerogel undergoes a series of chemical and physical modifications. Initially, the graphene oxide (GO) sheets are reduced to rGO to restore the conductive properties of graphene by removing oxygen-containing groups [33]. This reduction is typically achieved through chemical reduction methods using agents like hydrazine, ascorbic acid, or other environmentally friendly reducing agents [22].

Previous studies have studied graphene oxide-based aerogels with cysteamine modification for dye removal [38]. In this study, reduced Graphene Oxide (rGO) was successfully modified by functionalization with a thiol molecule (cysteamine) through a thiolene click chemistry reaction [39]. rGO has a higher surface area and a higher number of C=C bonds per unit surface area, which promotes the thiolene click reaction to retain cysteamine on the rGO surface compared to GO alone. The rGO-aerogel product of this process combines the remarkable properties of both aerogels and graphene: high surface area, mechanical strength and flexibility, thermal and electrical conductivity, chemical reactivity, and functionalization. Studies confirmed the formation of a mixture of open and closed ring structures based on amides, imides, and imines. The study utilized thiolene click chemistry for functionalizing rGO, with cysteamine chosen due to its advantages in covalently bonding with graphene and facilitating further functionalization. The novelty of this study is to examine the adsorption behavior of cations and less polar carbonaceous contaminants, which resulted in the production of reduced graphene oxide-cysteamine aerogel (rGO-aerogel). With its unique findings, the research demonstrated the immense potential of the rGO-aerogel for enhancing the removal of Ni^2+^ and MO from an aqueous solution, providing a solid foundation for further exploration and application of this material.

## 2. Experimental

### 2.1. Materials

Graphite powder (100% purity) was purchased from Fisher Scientific Company (Waltham, MA, USA). 2-aminoethanethiol (cysteamine) (C_2_H_7_NS) (98% purity) was supplied by TCI America (Portland, OR, USA) and used as received. Methyl orange (C_14_H_14_N_3_NaO_3_S) (98% purity) was purchased from Acros Organics (Geel, Belgium). Nickel chloride hexahydrate (NiCl_2_·6H_2_O) (98% purity) was purchased from Fisher Chemical (Pittsburgh, PA, USA). Ethanol (99.5% purity) was purchased from Fisher Chemical (Pittsburgh, PA, USA).

### 2.2. Methods

The GO nanosheets were synthesized according to the modified Hummer’s method as previously reported [40,41]. The rGO was synthesized by the reduction of GO from the prepared GO sheets. A suspension of GO (0.1 mg mL^−1^) was prepared by sonicating dried GO in distilled water. To its 100 mL, 100 mg of ascorbic acid was added. The pH of the medium was adjusted to ∼10 by adding NH_3_ solution to promote colloidal stability through electrostatic repulsion. The mixture was allowed to stir at 65 °C for 1 h. The formation of rGO was indicated by the color of the suspension changing from dark brown to black. Finally, the obtained product was centrifuged at 5000 rpm and washed with deionized water several times to remove impurities, if any were present. The obtained rGO was dried in a hot air oven at 100 °C overnight [42,43].

For the preparation of rGO-cysteamine hydrogel, the first 80 mg of cysteamine was mixed with 320 mg of rGO in 80 mL of deionized water (pH 5.8, conductivity = 7.6 μS cm^−1^) at a stirring rate of 200 rpm for 2 h at 25 °C. The solution was transferred to a high-pressure reactor for 4 h at 100 °C, ensuring a comprehensive reaction to obtain the hydrogel. The formed hydrogel was immersed in 25% (*v*/*v*) aqueous ethanol for 2 h to remove impurities. The rGO-cysteamine aerogel was obtained by freeze-drying, in which the sample was first frozen at −40 °C for 2 h and then dried by a vacuum freeze-dryer (−60 °C and 10 Pa) for 24 h. The schematic representation of the rGO-aerogel preparation can be seen in Figure 1.

### 2.3. Characterization of rGO and rGO-Aerogel

The rGO and rGO-aerogel were each individually characterized unless specified. The N_2_ Brunauer–Emmett–Teller (BET) specific surface area, pore volume, and pore sizes were determined using a N_2_ adsorption isotherm (NOVA 600 Anton paar instrument); each sample was degassed at 60.0 °C for 4 h. A JEOL EDS FE instrument was used to generate the scanning electron microscopy (SEM) (JSM-6010LA, JEOL (Tokyo, Japan), using the InTouch Scope software with version number 1.11) for surface morphology. The X-Ray photoelectron spectroscopy (XPS) data were acquired using EnviroESCA with MonoAl X-Ray source, with an emission current of 3 mA and at a voltage of 14 kV. A Thermo Scientific Flash 2000 organic elemental analyzer was used to determine the C, H, N, and S values. A PerkinElmer-S11 system (Perkin Elmer, Waltham, MA, USA) was used for the Thermogravimetric Analysis (TGA) to study thermal stability. The thermal flow properties of the samples were analyzed with a Differential Scanning Calorimeter (DSC) using the Mettler Toledo DSC-3 system (Mettler Toledo, Columbus, OH, USA). The functional groups of rGO and the rGO-aerogel were determined by a Thermo Scientific Nicolet instrument through Fourier transform infrared (FTIR) spectroscopy (Thermo Fisher Scientific, Houston, TX, USA). UV-Vis spectroscopy was used to determine the remaining concentrations of MO in the adsorption studies (Shimadzu UV-1800, Shimadzu, Kyoto, Japan). The solution’s remaining Ni^2+^ concentrations were analyzed with Anton Paar instruments and Atomic Adsorption Spectroscopy (AAS). A 400 MHz Bruker Ascend EVO 400 instrument (Bruker Corporation, Billerica, MA, USA) was used for Nuclear Magnetic spectra (NMR) analysis, and the Topspin v. 3.6.3 software was used for the NMR data analysis.

### 2.4. Adsorption Experiments

All the adsorption experiments were performed in 50 mL polypropylene vials. In total, 0.1 g/L adsorbent doses were equilibrated with MO and Ni^2+^ solutions for 4 h at 200 rpm. The Ni^2+^ filtrates were filtered after equilibration using Whatman grade 1 cellulose filter papers (11 μm pore size), followed by measuring the remaining concentration using an Anton Paar atomic absorption spectrometer. The MO supernatants were separated from 3.5 mL pipettes, and the remaining concentrations were measured using a Shimadzu UV-Vis spectrophotometer. The adsorption capacities and removal percentages of MO and Ni^2+^ were calculated as follows (Equations (1) and (2)):(1)$$Q_{\mathrm{Eq}}=\frac{V\left(C_{\mathrm{Initial}}-C_{\mathrm{Eq}}\right)}{m}$$
(2)$$\mathrm{Removal}\;\mathrm{Percentage}\;\left(\%\right)=\frac{C_{\mathrm{Initial}}-C_{\mathrm{Eq}}}{C_{\mathrm{Initial}}}\times 100\%$$

Here, C_Initial_ and C_Eq_ represent the initial and equilibrium concentrations of MO and Ni^2+^ in mg L^−1^, respectively. V and m are the volume of adsorbate and mass of adsorbent in L and g, respectively.

## 3. Results and Discussion

### 3.1. Characterization

The microstructure and morphology of the rGO and rGO-aerogel were examined using Field emission-Scanning electron microscopy (FE-SEM) at two different magnifications (×250 and ×2500). The rGO-aerogel revealed interesting findings in the SEM images (Figure 2a–d). The rGO-aerogel displayed a rippled, flaky, silk-like wave structure in comparison to the rGO, with an aggregated appearance with fewer wrinkles, which could be due to the oxygen reduction process during production [44,45]. The rGO-aerogel displays more micro- and mesopores on the surface of interconnected rGO layers; the stacking of rGO layers in the aerogel is effectively decreased, mainly due to the chemical modification and crosslinking of rGO with cysteamine. The synthesized aerogel in between the graphene layers increased the hydrophilicity of the material. The 3D network structure of the rGO-aerogel can be observed, contrasting with the collapsed structure of the rGO. Furthermore, after the addition of cysteamine, the high-magnification SEM image showed that the pore size of the rGO-aerogel increased and the walls of the pores thinned, suggesting the decrease in the re-stacking and steric hindrance of the exfoliated rGO. These results indicate that cysteamine successfully synthesized aerogel in between the graphene layers of rGO, and the three-dimensional, porous, and exposed carbonaceous surface of graphene layers in the rGO-aerogel is highly desirable for organic pollutants and heavy metal cleanup [46,47].

In order to conduct a thorough examination of the porous structures of the samples, N_2_ adsorption–desorption measurements were performed with the BET instrument (Figure 2e), revealing a significant finding. The specific surface areas of the rGO saw a drastic increase, up to ~10.7 times, from 29.3 m^2^ g^−1^ to 313.3 m^2^ g^−1^ for the rGO-aerogel (Table 1), a finding that supports the high porosity (Figure 2a–d). This increase is largely attributed to the introduction of cysteamine, which played a pivotal role in significantly increasing the specific surface area of the obtained rGO-aerogel. The corresponding pore size distribution curves are depicted in Figure 2f, confirming that the hierarchical porous network structure of the rGO-aerogel exhibited excellent micro and mesopores compared to the rGO, which is consistent with the porous structure shown in the SEM images in Figure 2c,d. The number of micropores in the rGO-aerogel shows a surge compared with that of the rGO. The pores in the 5–40 nm mesoporous region uniformly increased. It indicates that the cysteamine modification generates abundant pore structures in the resultant rGO-aerogel. The higher micropore and mesopore volumes of the rGO-aerogel are advantageous for pollutant adsorption, as mesopores provide rapid ion transport pathways and micropores offer a large surface area to accommodate pollutants.

The FT-IR spectra of the rGO and rGO-aerogel are presented in Figure 2g, demonstrating the successful synthesis of cysteamine aerogel with rGO. The rGO-aerogel displayed intense peaks at 2570, 3412, and 1620 cm^−1^ that corresponds to -S-H stretching, -N-H stretching, and amide -C=O stretching, respectively. Further, the presence of C-N bending and H-C-H bending at 1189 and 1411 cm^−1^, respectively, in the fingerprint region also confirmed the presence of cysteamine on the rGO surface.

Carbon, hydrogen, nitrogen, and sulfur elemental analyses were performed for both samples using combustion analysis (Table 1). The oxygen contents were calculated as $O\left[\%\right]=100\left[\%\right]-\left(\mathrm{Ash}\left[\%\right]+C\left[\%\right]+H\left[\%\right]+N\left[\%\right]+S\left[\%\right]\right)$ where the ash percentages were 0% for both the rGO and rGO-aerogel materials. The resultant sulfur in the rGO (0.81%) increased to 1.91%, confirming the cysteamine modification in the formation of the rGO-aerogel. It can be observed that the surface modification of the rGO was accompanied by an increase in the carbon content and a decrease in the oxygen content of the rGO-aerogel.

NMR spectra analysis was conducted with approximately 15–50 mg of Heavy Water, Deuterium Oxide (D_2_O) reaction mix dissolved in a glass NMR test tube in 0.5 mL deuterated chloroform (CCl_3_D). For ^1^H NMR, 32 scans were obtained per sample at a free induction decay (FID) resolution of 0.25 Hz and at 3.99 s acquisition time. For ^13^C NMR, 1024 scans were obtained per sample at an FID resolution of 0.72 Hz and 1.37 s acquisition time. The strong peak at 3.3 ppm of NMR could be due to the presence of the S-H functional group from cysteamine on the rGO-aerogel (Figure 3) [48].

The TGA curve, with its derivative, is depicted in Figure 2h. Under a nitrogen atmosphere, the rGO-aerogel samples were heated from 25 to 800 °C at 10 °C min^−1^. The initial weight loss observed from 40 to 100 °C was due to either water loss or other volatile substances inside the aerogel’s pores. The second weight loss between 280 and 300 °C corresponds to the decomposition of oxygen functional groups such as CO_2_ and/or H_2_O. Moreover, it was found that the third step of weight loss occurred during 280–580 °C, which is attributed to the removal of CO_2_/H_2_O from some more stable oxygen functional groups, along with the -OH, -COOH, and -C-OH groups [49,50]. The peak around ~525 °C in the derivative is related to the thermal decomposition of the rGO-aerogel structure. The reduction of oxygen functional groups in the structure of the rGO-aerogel, whose functional groups are easily thermally degraded to convert into other low molecular compounds such as CO_2_ or H_2_O, has improved the thermal stability of this rGO-aerogel [46].

The DSC analysis was performed during the heating and cooling of the samples of the rGO and rGO-aerogel at a heating rate of 5 °C min^−1^, and the temperature range was set to 25–250 °C (Figure 4a,b). The heating/cooling process was conducted under a nitrogen atmosphere. The phase change temperatures varied slightly for both materials, but the latent heat decreased due to the existence of the porous structure [51].

The XPS analysis plays a pivotal role in providing essential information about the structure and composition of these materials’ chemical composition and surface properties (Figure 4c–f). The peaks attributed to different elements and functional groups reveal crucial details about their structure and composition. The presence of peaks attributed to C1s, N1s, and O1s indicates the presence of carbon, nitrogen, and oxygen in both the rGO and rGO-aerogel. These elements are fundamental components of organic materials and are essential for understanding the chemical nature of these substances. The S2p peak on the rGO-aerogel structure indicates the presence of sulfur in the form of cysteamine [46,52,53].

### 3.2. Adsorption Studies

#### 3.2.1. Effect of pH on Adsorption

The point of zero charge (PZC) experiments were performed in a 0.01 M NaCl solution equilibrated with an adsorbent dose of 0.1 g L^−1^ at 25 °C. The final and initial pH of the solutions were measured using a Fischer Scientific pH meter. The PZC of the rGO-aerogel is 6.8, where the rGO-aerogel has a net positive charge below pH 6.8 and, vice versa, a net negative charge above pH 6.8 (Figure 5a). The rGO-aerogel shows a slightly alkaline nature in a de-ionized water matrix (pH_DI water_ 6.5). The pH level of aqueous solutions significantly influences the adsorption process of contaminants on adsorbents.

Thus, an investigation into the impact of pH on MO adsorption performance was carried out within the pH range of 2.0 to 10.0, revealing some intriguing findings at 20 mg L^−1^ initial MO and Ni^2+^ concentration with a 0.1 g L^−1^ adsorbent dose at 25 °C. MO has a pK_a_ value of 3.5, where the anionic form exhibits above pH > pK_a_ and the net zero charge species exhibits at pH < pK_a_ (Figure 5c). The maximum MO uptake on the rGO-aerogel occurred in a wide pH range, from the initial pH of 4.0 to a pH of 7.0 with a removal percentage of ~100%. Over the initial solution pH range of 2.0–10.0, the absorption was ~80% or more, demonstrating the excellent pH window provided by this adsorbent.

Ni^2+^ exists as Ni^2+^, Ni (OH)^+^, Ni (OH)_2_, Ni (OH)_3_^−^, and Ni (OH)_4_^2−^ at the pH ranges of <7.8, 7.8–10.9, 8.3–13.1, >9.3, and >11.8, respectively (Figure 5b). The Ni^2+^ uptake from the rGO-aerogel was low until the initial pH of 4 due to the higher competition of H^+^ towards the rGO-aerogel. For Ni^2+^, maximum uptake on the rGO-aerogel occurred at pH 5.0. However, Ni^2+^ adsorption was reduced above pH 6.0 due to the reduction in the net positive charge of Ni^2+^ which forms ionic interactions with the rGO-aerogel.

The pH trend of Ni^2+^ is quite opposite to MO, with an uptake of 20% or more over the pH range of 2.0–10.0. This characteristic is primarily attributed to the specific synthesis procedure employed. Notably, the higher surface area of the rGO-aerogel facilitates the presence of more active sites for the adsorption of MO and Ni^2+^, consequently leading to an accelerated adsorption rate. Moreover, a more significant concentration gradient between the solution and the adsorbent surface can contribute to an initially faster adsorption rate. It is important to note that pH level variations can influence the adsorption rate by modifying the electrostatic interactions between contaminants and the adsorbent surface. Additionally, the agitation or stirring of the solution can play a significant role in improving mass transfer and the contact between contaminants and the adsorbent surface. Furthermore, smaller particle sizes and higher porosity can effectively augment the available surface area for adsorption, facilitating swifter MO and Ni^2+^ adsorption; however, these parameters were not studied in this work.

Below pH 8, cysteamine exists as ^+^NH_3_-CH_2_-CH_2_-SH, where cysteamine’s pK_a_ values are 8.19 and 10.75, respectively, for –SH and –NH_3_^+^ groups [54]. Ni^2+^ can form ion–dipole interactions with –SH groups and MO can form ionic interactions between positively charged –NH_3_^+^ groups on cysteamine and negatively charged –SO_3_^−^ groups on MO. These additional interactions improve the capacities of Ni^2+^ and MO on the rGO-aerogel.

#### 3.2.2. Adsorption Kinetics Studies

In general, the interaction between the contaminant and the adsorbent surface determines the adsorption of MO and Ni^2+^. In this study, the adsorption experimental data for MO and Ni^2+^ from water were fitted using pseudo-first- and pseudo-second-order kinetics, instead of other adsorption kinetic models that are commonly employed to fit the experimental data. The nonlinear form of the pseudo-first- and pseudo-second-order kinetic models’ adsorbent capacity-based rate expression is formulated by Equations (3) and (4), respectively.

The pseudo-first-order kinetic model equation [55,56]
(3)$$q_{t}=q_{e\;}\left(1-e^{-k_{1}t}\right)$$

The pseudo-second-order kinetic model equation [55,56]
(4)$$q_{t}=k_{2}\times q_{e}^{2}\times \frac{t}{1+k_{2}\times q_{e}\times t}$$
where q_e_ and q_t_ are the amount of dye adsorbed at equilibrium (mg g^−1^) and at time t (mg g^−1^), respectively, and k_1_ and k_2_ are the pseudo-first-order rate constant (min^−1^) and pseudo-second-order rate constant (g mg^−1^ min^−1^), respectively.

The adsorption kinetics of MO and Ni^2+^ with initial 20 mg L^−1^ concentrations at 25 °C with a dose of 0.1 g L^−1^ were analyzed using pseudo-first-order and pseudo-second-order kinetic models. These models help understand the adsorption process’ mechanism and rate-controlling steps, which are crucial for the design of efficient adsorption processes. Figure 6a,b depicts the experimental data and the fitting results with the corresponding parameters in the inlet.

The rGO-aerogel exhibited remarkable performance, achieving a 95% removal of MO within the first 15 min, and reaching equilibrium with 100% removal within 60 min. The R^2^ value for the pseudo-second-order kinetic model is 0.98, which is higher than the R^2^ value for the pseudo-first-order kinetic model (0.79). This indicates that the pseudo-second-order kinetic model better describes the adsorption kinetics of MO. The superior fit of the pseudo-second-order model suggests that the adsorption of MO onto the rGO-aerogel involves chemisorption, likely through electrostatic interactions and possibly π-π interactions between MO molecules and the aerogel surface. The rapid adsorption of Ni^2+^ in the initial stages can be attributed to the abundance of available active sites on the surface of the rGO-aerogel, while MO sites are readily accessible to the dye molecules, facilitating swift adsorption through physical or chemical interactions. The process is driven by strong electrostatic interactions between the negatively charged MO molecules and the positively charged sites on the rGO-aerogel surface. This strong attraction leads to the quick occupation of available sites, causing a steep increase in adsorption capacity. For Ni^2+^, the R^2^ value for the pseudo-first-order kinetic model is 0.98, which is higher than that for the pseudo-second-order kinetic model (0.97). For Ni^2+^, the maximum removal capacity of 75% was attained at 360 min. Thus, the pseudo-first-order kinetic model better describes the adsorption kinetics of Ni^2+^.

The adsorption of MO and Ni^2+^ onto the rGO-aerogel is likely driven by strong interactions between the negatively charged dye molecules and positively charged adsorption sites on the rGO-aerogel. The pseudo-second-order kinetic model’s better fit implies that the adsorption process involves the π-π interactions between MO and the ionic interactions between Ni^2+^ with the rGO-aerogel’s surface, indicating chemisorption. The adsorption rate is more dependent on the availability of free adsorption sites rather than the interaction strength in this scenario, leading to a fit that favors the pseudo-second-order model. These findings have implications for the design of adsorption processes as they highlight the importance of considering the adsorbate–adsorbent interactions and the availability of adsorption sites.

#### 3.2.3. Adsorption Isotherm Studies

The adsorption isotherm, a key concept in understanding the distribution of adsorbate molecules between solid–liquid phases at equilibrium, is further elucidated by two prominent isotherm models, the Langmuir model and the Freundlich model. These models, play a crucial role in explaining the equilibrium characteristics of adsorption. The Langmuir isotherm model describes the monolayer adsorption process on uniform adsorption sites and is expressed by Equation (5) [57], whereas the Freundlich isotherm model describes the multilayer adsorption approach and is expressed by Equation (6) [58].
(5)$$q_{e}=\frac{q_{\max}K_{L}C_{e}}{1+K_{L}C_{e}}$$
(6)$$q_{e}=K_{F}C_{e}^{1/n}$$
where q_e_ is the quantity of target contaminant adsorbed per unit weight of adsorbent at equilibrium concentration (mg g^−1^), C_e_ is the concentration of equilibrium (mg L^−1^), q_max_ is the Langmuir maximum adsorption capacity (mg g^−1^), K_L_ is the Langmuir constant (L mg^−1^), K_F_ is the Freundlich maximum adsorption capacity (mg g^−1^), and 1/n is the adsorption intensity of the system.

Our investigation into the isotherms involved calculating the experimental data under optimal conditions, which included varying the initial pollutant concentrations from 10 to 100 mg L^−1^ and maintaining temperatures at 10, 25, and 40 °C. The isotherm parameters, a key outcome of our study, were rigorously predicted by the suggested models at these three temperatures. The parameters are detailed in Table 2, and the corresponding fit is visually represented in Figure 7a–d. The Langmuir isotherm model best fits the data for both MO and Ni^2+^, indicating monolayer adsorption onto a surface with a finite number of identical sites. The maximum Langmuir adsorption capacities were 542.6 mg g^−1^ for MO and 150.6 mg g^−1^ for Ni^2+^ at 25 °C. Importantly, the adsorption process was identified as chemisorptive, a significant finding that suggests that the pollutants form strong chemical bonds with the active sites on the rGO-aerogel, providing a deeper understanding of the interaction between the pollutants and the rGO-aerogel. Table 3 presents a comparison of the adsorption capacity of different modified aerogel adsorbents towards MO and Ni^2+^ in aqueous solutions documented in the recent literature.

## 4. Conclusions

This study represents a pioneering effort in water purification by exploring the potential of modified reduced graphene oxide, specifically enhanced with cysteamine, a novel material (rGO-aerogel), for removing organic contaminants and heavy metals from wastewater streams. The synthesis and characterization of this modified rGO-aerogel were carefully investigated utilizing a range of analytical techniques, such as elemental analysis, FE-SEM, BET, FTIR, TGA, DSC, XPS, and NMR. The successful interaction between the thiol in cysteamine and the double bonds on the surface of the rGO, as well as the pyrazolic structures created as a result of reduction, was confirmed by the existence of the C-S bond in the X-Ray photoelectron spectrum. These techniques gave important insights into the surface chemistry and structural alterations of the rGO-aerogel, which are essential for comprehending its improved adsorption capabilities. The maximum removal of nickel ions (Ni^2+^) and methyl orange (MO) was seen on the rGO-aerogel at pH ranges of 4–7 and pH 5, respectively. We examined the adsorption kinetics of MO and Ni^2+^ onto the rGO-aerogel. The adsorption rate for MO was remarkably high: it reached equilibrium at 100% removal in 60 min after obtaining 95% removal in the first 15 min. While the notable, but longer, adsorption of Ni^2+^ indicates the rGO-aerogel’s suitability for heavy metal remediation, the quick and effective adsorption of MO emphasizes this material’s promise for treating organic contaminants. The Langmuir isotherm of MO indicated a maximum adsorption capacity of 542.6 mg g^−1^, and a Ni^2+^ content of 150.6 mg g^−1^ at 25 °C. The study provides a comprehensive understanding of the adsorption properties and kinetics of rGO-aerogel and offers a promising solution to the global water pollution crisis. rGO-aerogel could be a potential material for treating heavy metal- and organic dye-contaminated wastewater with the adsorption data and characteristic features.

## Figures and Tables

**Figure 1 nanomaterials-14-01708-f001:**
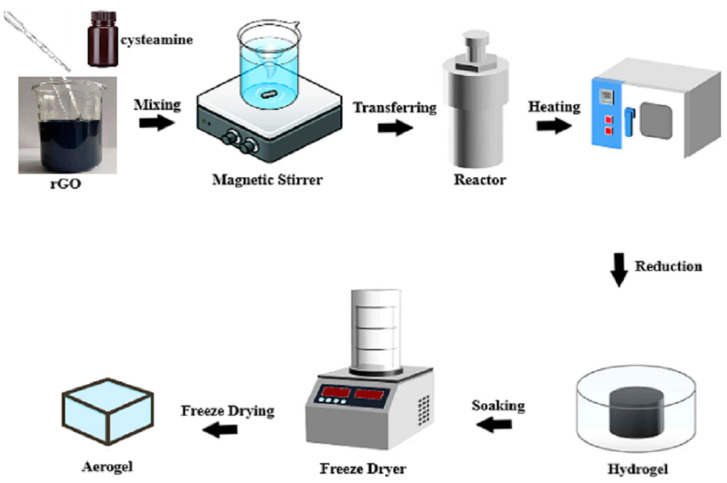
Schematic representation of rGO-aerogel preparation.

**Figure 2 nanomaterials-14-01708-f002:**
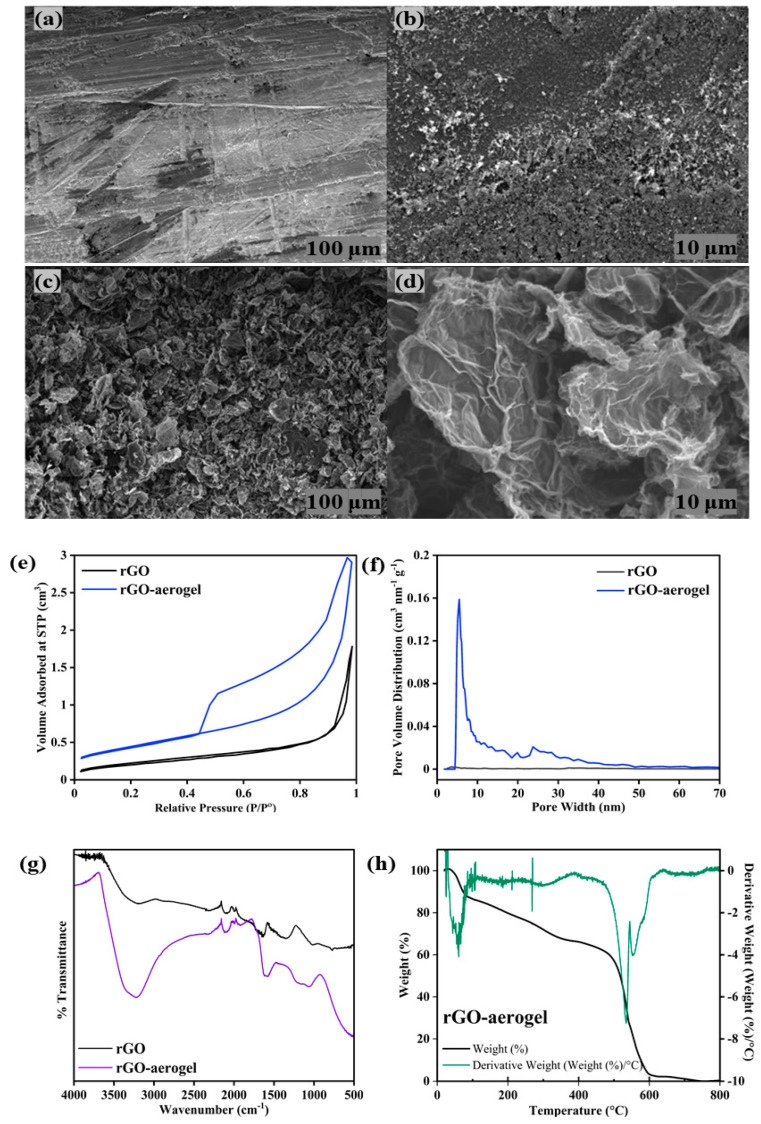
FE-SEM micrographs of (**a**,**b**) rGO and (**c**,**d**) rGO-aerogel. (**e**,**f**) Nitrogen adsorption–desorption isotherms and the corresponding average pore size distribution of rGO and rGO-aerogel. (**g**) FT-IR Spectra of rGO and rGO-aerogel and (**h**) TGA/DTG graphs of rGO-aerogel.

**Figure 3 nanomaterials-14-01708-f003:**
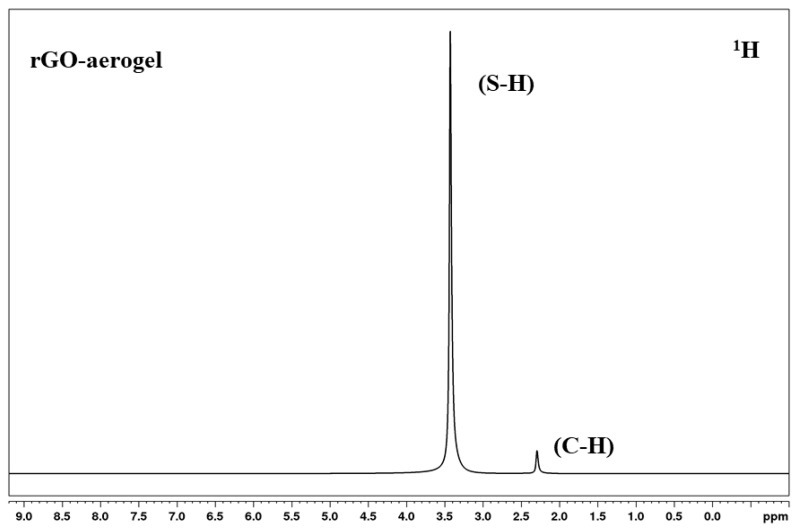
NMR spectra of rGO-aerogel.

**Figure 4 nanomaterials-14-01708-f004:**
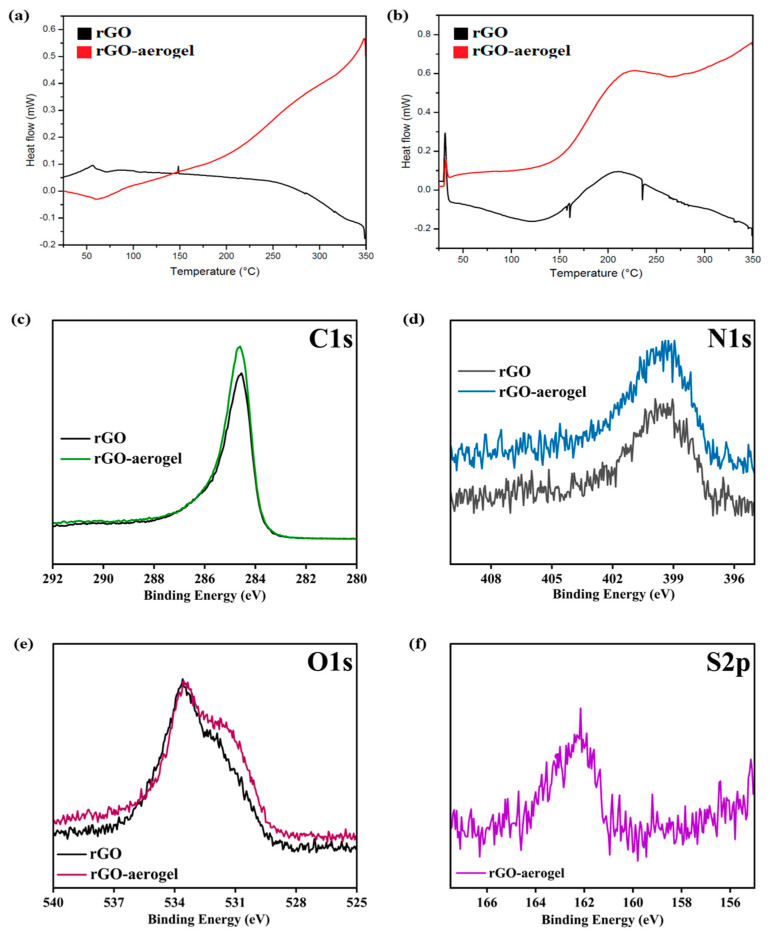
DSC profile of rGO and rGO-aerogel (**a**,**b**) heating and cooling scan. (**c**–**f**) High resolution (HR) C1s, N1s, O1s, and S2p XPS spectra patterns of rGO and rGO-aerogel.

**Figure 5 nanomaterials-14-01708-f005:**
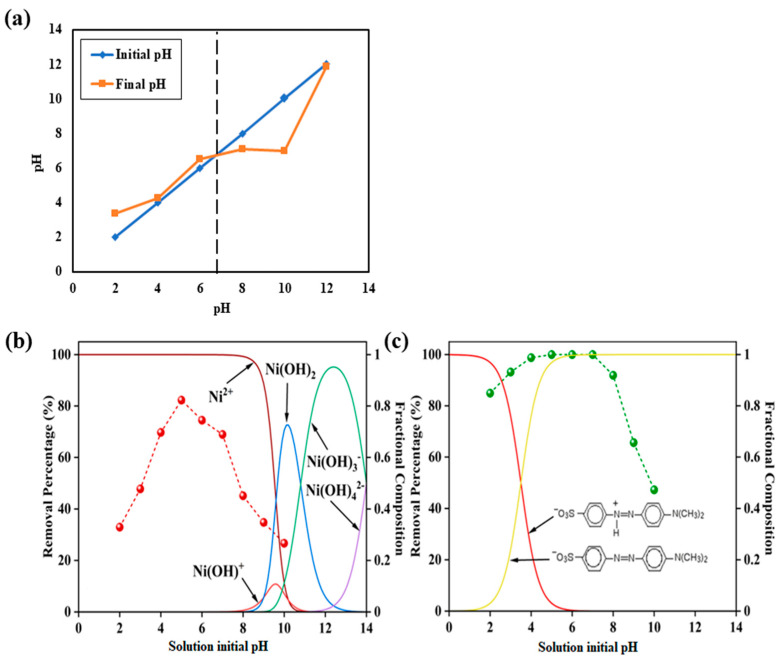
(**a**) Point of zero charge of rGO-aerogel. (**b**,**c**) Effect of pH on % removal of Ni^2+^ and MO.

**Figure 6 nanomaterials-14-01708-f006:**
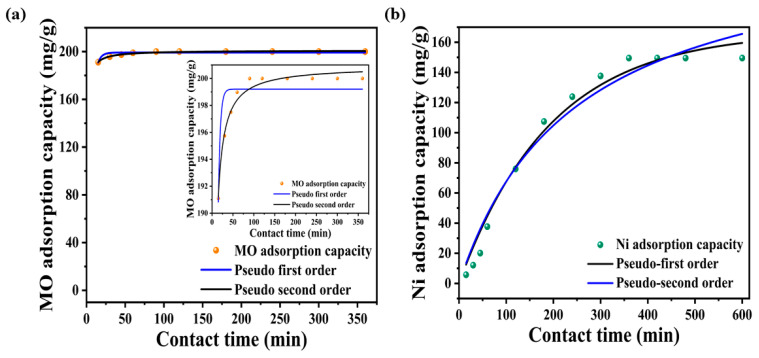
Adsorption kinetics plots fitted with pseudo-first-order and pseudo-second-order models for (**a**) MO and (**b**) Ni^2+^.

**Figure 7 nanomaterials-14-01708-f007:**
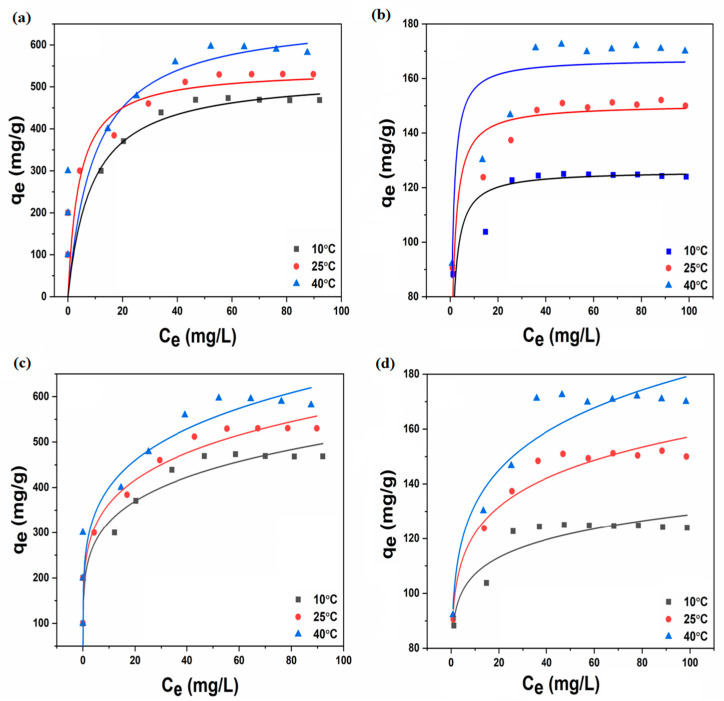
Adsorption isotherm plots fitted with (**a**,**b**) Langmuir model for MO and Ni^2+^, and (**c**,**d**) Freundlich model for MO and Ni^2+^.

**Table 1 nanomaterials-14-01708-t001:** Elemental analysis and surface area of rGO and rGO-aerogel.

Sample	Elemental Analysis					Surface Area (m^2^/g)
	%C	%H	%N	%S	%O	
rGO	77.52	1.53	4.20	0.81	15.94	29.3
rGO-aerogel	74.05	0.77	4.15	1.91	19.10	313.3

**Table 2 nanomaterials-14-01708-t002:** Isotherm parameters of MO and Ni^2+^ adsorption.

Isotherm Models	MO			Ni^2+^		
Temperature (°C)	10	25	40	10	25	40
**Langmuir**						
q_m_ (mg g^−1^)	528.70	542.59	670.63	126.19	150.60	167.25
K_L_ (L mg^−1^)	0.11	0.24	0.10	0.29	0.25	0.44
R^2^	0.931	0.949	0.945	0.893	0.944	0.924
**Freundlich**						
K_F_ (mg g^−1^)	207.10	232.66	251.19	88.92	94.57	97.67
n	5.17	5.14	4.95	12.43	9.05	7.57
R^2^	0.855	0.894	0.901	0.871	0.944	0.92

**Table 3 nanomaterials-14-01708-t003:** Adsorption capacity of different modified aerogel adsorbents towards MO and Ni ions in aqueous solutions.

Adsorbent	Adsorbate	q_m_ (mg g^−1^)	Reference
Ti_3_C_2_T_X_ MXene aerogel	MO	106.3	[59]
Graphene aerogel	MO	3059.2	[60]
Chitosan aerogel	MO	179.2	[61]
RC/PCA aerogel	MO	980.4	[62]
TDB aerogel	MO	488.2	[63]
Graphene/Chitosan aerogel	MO	543.4	[64]
MOF-199@AFGO/CS aerogel	MO	412.0	[65]
KGM/GO/ZIF-67 aerogel	MO	130.2	[66]
RC/CSA aerogel	MO	742.7	[67]
rGO-aerogel	MO	542.6	This study
Zn/Ni carbon aerogel	Ni^2+^	194.2	[68]
Chitosan/Silica aerogel	Ni^2+^	58.8	[69]
Nano hydroxyapatite	Ni^2+^	40.0	[70]
Cellulose-based adsorbent	Ni^2+^	295.9	[71]
GO/paper	Ni^2+^	29.0	[72]
Activated carbon	Ni^2+^	50.5	[73]
rGO-aerogel	Ni^2+^	150.6	This study

## Data Availability

Data are contained within the article.

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
