# Peer review of "Harnessing Novel Reduced Graphene Oxide-Based Aerogel for Efficient Organic Contaminant and Heavy Metal Removal in Aqueous Environments"

_nanomaterials, 2024, doi:10.3390/nano14211708_

Round 1

Reviewer 1 Report

Comments and Suggestions for Authors

The manuscript presents an innovative approach to the reduction of graphene oxide (rGO) aerogels using modified Eder, as well as an in-depth examination of the underlying mechanisms of contaminant removal through the lens of adsorption kinetics, equilibrium, and isothermal studies.It is interesting, but there are still some problems that need to be improved.

The following is what I think needs to be revised accordingly:

1. The manuscript contains numerous typographical, spelling, and grammatical errors that require revision and attention. Some important review DOI: 10.1016/j.cej.2024.149860 might be included and expanded in the introduction.

2. The highlights are too scattered. Could you please concentrate them into 3-4 subheadings?

3. The formulas and their serial numbers are inconsistent in format, please correct them.

4. Please optimize Table 1 to leave significant spacing between samples and numbers, and between numbers.

5. Please note that in Table 2, the "2+" for Ni ions is not superscripted.  Please double-check the entire text and correct it.

6. Why does the carbon content of modified rGO-aerogels increase and the oxygen content decrease? Please provide an explanation.

7. The principle of adsorption of rGO-aerogels is not explained in the manuscript, please add.

8. Whether the practicality of the aerogel is analyzed and whether real samples are used for experiments.

9. In the notes to Figure 5, the font sizes of Ni and MO do not match the font sizes of the surrounding text. Please double-check the entire text and correct it.

10. The manuscript was examined and it was determined that no 3.2.3 was present. Therefore, the serial number for the adsorption isotherm study should be 3.2.3.

11. Should this aerogel be compared with other similar materials, please show them in a table?

Comments on the Quality of English Language

Moderate editing of English language required.

Reviewer 2 Report

Comments and Suggestions for Authors

This paper investigates an innovative methodology utilizing modified reduced graphene oxide (rGO) aerogel for the adsorption of waterborne contaminants, thereby enhancing environmental remediation efforts. Focusing on this type of aerogel, the study offers a theoretical framework elucidating the mechanisms by which the material adsorbs pollutants in aqueous environments through various analytical techniques. While the content and logic of this paper are comprehensive, certain issues remain to be addressed, it is deemed suitable for publication after minor revisions.

1.In section 3.1, the sentence "The synthesized aerogel..." and "...superhydrophobic structures of aerogel..." It is mentioned that aerogel improves the hydrophilicity of the materials and the superhydrophobic structure of the aerogel. This conclusion needs to be verified by additional experiments.

2.What causes significant changes in the morphology and structure of graphene oxide aerogels after the addition of cysteamine, leading to increased porosity? How does graphene oxide crosslink with cysteamine?

3.XPS map has no data processing, and the existing data should be processed simply.

4.As mentioned in the abstract, the structural stability and extensibility of aerogel were evaluated. However, there is no detailed description of these two items in the article, please add complete relevant information.
